# Brief daily functional strength training to improve functional performance in older adults with mobility disability: A randomized trial

Smita Dandekar[1], Jordan Kurth[2], Yimeng Shang[3], Jonathan G. Stine[4,5], Matthew A. Ladwig[6], David E. Conroy[7], Kathryn H. Schmitz[8], Liza S. Rovniak[2,5], Matthew Silvis[9], Margaret Danilovich[10], Noel Ballentine[2], Natalia Pierwola-Gawin[2], Shouhao Zhou[3], Christopher Sciamanna[2,5]*

1 Department of Pediatrics, Division of Hematology and Oncology, Penn State Health Golisano, Children's Hospital, Hershey, Pennsylvania, United States of America, 2 Department of Medicine, Division of General Internal Medicine, Penn State College of Medicine, Hershey, Pennsylvania, United States of America, 3 Department of Public Health Sciences, Division of Biostatistics and Bioinformatics, Penn State College of Medicine, Hershey, Pennsylvania, United States of America, 4 Department of Medicine, Division of Gastroenterology and Hepatology, Penn State Health-Milton S. Hershey Medical Center, Hershey, Pennsylvania, United States of America, 5 Department of Public Health Sciences, Penn State College of Medicine, Hershey, Pennsylvania, United States of America, 6 Department of Biological Sciences and Integrative Physiology and Health Sciences Center, Purdue University Northwest, Hammond, Indiana, United States of America, 7 School of Kinesiology, University of Michigan, Ann Arbor, Michigan, United States of America, 8 Department of Medicine, Division of Hematology and Oncology, University of Pittsburgh School of Medicine, Pittsburgh, Pennsylvania, United States of America, 9 Departments of Orthopedics and Rehabilitation and Family and Community Medicine, Pennsylvania State Health-Milton S. Hershey Medical Center, Hershey, Pennsylvania, United States of America, 10 Department of Physical Therapy and Human Movement Science, Northwestern University, Feinberg School of Medicine, Chicago, Illinois, United States of America

* cns10@psu.edu

## Abstract

### Objectives

Mobility disability is associated with functional decline in older adults. Resistance training (RT) improves mobility disability, but adherence to national RT guidelines is poor. We evaluated the effects of a 12-week brief, home-based functional RT program, FAST (*Functional Activity Strength Training)*-2, on adherence and functional impairment in older, inactive adults ≥ 65 years of age, with pre-existing walking difficulty.

### Methods

Eligible older adults were randomized using stratified assignment based on biological sex and age (65–72 and 73+) to either the FAST-2 intervention involving a 4-minute daily workout of four exercises lasting 30 seconds each or the delayed treatment control condition. Video coaching at baseline and at weeks 2, 4 and 8, provided feedback on exercise form, modifications and progression. Daily email reminders were sent for workout completion, and to report exercise performance and rate perceived

**Data availability statement:** The data associated with this study is available via Penn State Data Commons at https://doi.org/10.26208/WVV2-TX77.

**Funding:** The author(s) received no specific funding for this work.

**Competing interests:** The author(s) declared the following potential conflicts of interest with respect to the research, authorship, and/or publication of this article: Dr. Sciamanna is part-owner of BandUp, Inc. and Play Fitness, LLC, formed to test the viability of various business models to disseminate findings from exercise studies. Dr. Jonathan Stine receives or has received research support from Astra Zeneca, Galectin, Kowa, Noom, Inc, Novo Nordisk, Regeneron and Zydus Therapeutics. Dr. Stine consults for Novo Nordisk and is on an advisory board for Madrigal. No other authors have competing interests. This does not alter our adherence to PLOS ONE policies on sharing data and materials.

exertion. Performance and adherence feedback were emailed biweekly. Functional performance was measured by video using the Five-Times Sit-to-Stand (FTSTS) test, One-Legged Stance Test (OLST) and the 30-second chair stand test at baseline and at weeks 6 and 12.

## Results

Ninety-seven participants were randomized to either the FAST-2 treatment intervention (n = 44) or the delayed treatment control condition (n = 53). The linear mixed-effect model showed the intervention group decreased the FTSTS by 2.3 seconds (95% CI: 0.5–4.1, p = 0.01), increased OLST by 3.6 seconds (95% CI: 0.6–6.5, p = 0.02) and increased the number of chair stands by 4.2 repetitions (95% CI: 2.8–5.7, p < 0.001) more than the control group over 12 weeks. Intervention participants completed the workout 81% of the days. No significant adverse events were reported.

## Conclusion

The 12-week FAST-2 intervention, including only 60-seconds of lower extremity exercises in older individuals with pre-existing walking difficulty, yielded improvement in functional performance.

## Trial registration

ClinicalTrials.gov: ID NCT05697497
Study Details | NCT05697497 | Functional Activity Strength Training | ClinicalTrials.gov

---

## Introduction

One in four older adults, the fastest growing demographic group in the US [1], reports "serious difficulty walking or climbing stairs", referred to as "mobility disability". [2,3] In qualitative studies among older adults, participants have stated that mobility disability "deprives you of your identity", "prevents me from doing many things I used to enjoy, like walking", "affects my day to day life and "I can't do very much work on my own". [4] Those with mobility disability are 8.7 times more likely to die, incur an additional $10,000 each year in health care costs and are 13–36 times more likely to transition to a nursing home in the near future [5–7].

Although resistance training (RT) improves mobility disability [8], too few older adults do it. Systematic reviews observe that 6 months of RT increases strength by 50% in older adults [9,10], which improves mobility (Cohen's d = 0.61, 5 trials) [8]. Despite these benefits, fewer than 20% of older adults meet national guidelines for doing RT twice per week. [11–13] Even when programs are provided at no extra cost by health insurance plans (e.g., SilverSneakers), fewer than 30% of older adults participate [14] and those that do participate rarely attend (average < 20 visits/year).

[15,16] What remains unknown is how to create RT options for older adults that improve physical function and that most are willing to do.

One approach that may increase the use of RT, though it remains untested, is making RT sessions shorter. A narrative review exploring time efficient ways to structure strength training suggested very short and frequent workout sessions as a viable alternative for individuals reluctant to engage in longer training sessions. [17] Our team has observed that 84% of older adults with difficulty walking preferred doing RT 5 minutes per day versus the traditional 45 minutes three-times weekly in part due to physical limitations and pain. [18,19] These briefer RT sessions are also supported by evidence that aerobic exercise high-intensity interval training leads to large increases in fitness in as little as 3 minutes per week [20,21], while systematic reviews observe that most of the strength gains are from the first few sets each week. [22,23] Additionally, studies have shown that more than a minimal volume of exercise can significantly help prevent numerous chronic diseases and dementia.] These prior studies and guidelines suggest that longer programs that include RT may not always be necessary for health benefits and might discourage participation among people with mobility challenges. [24] Our goal, therefore, was to design a brief RT program that could improve physical function among older adults.

This project began in 2020 with the goal of identifying the briefest dose of RT that could improve mobility disability among older adults. The project was called FAST (Functional Activity Strength Training) and FAST was augmented with goal-setting, rarely used in RT studies [25], for the number of additional repetitions participants should be able to do over time. In the first study (FAST-1), 24 healthy older adults were prescribed 30 seconds of squats and push-ups each day and given no personal supervision. Over six months, participants performed the exercises on 73% of days and showed large increases in squat performance (+6.2 repetitions, Cohen's d > 1.0). [26] This study, however, enrolled a small number of healthy older adults, evaluated only self-reported outcomes and lacked a control group.

In this study (FAST2) we set out to test whether FAST can, in 4 minutes of RT per day, improve physical function among older adults with mobility disability. We hypothesized that participants randomized to the FAST-2 treatment intervention would improve measures of lower extremity performance that are strongly associated with future disability, in 12 weeks, compared to those randomized to the delayed treatment control. These data could help clarify if brief RT programs could remove barriers to disseminating RT and improve older adults' physical function.

## Materials and methods

### Study design

FAST-2 was a two-group, 12-week, delayed-treatment randomized trial. This duration was selected as 12 weeks is the typical length of exercise trials. Data suggests that 12 weeks is long enough to allow for physiological changes to occur, both the neural and muscular changes associated with strength training. Initial trials (i.e., those testing a new intervention) typically last 12 weeks to balance participant burden with the ability to test the efficacy of the program. Then, longer trials can be conducted if the 12-week intervention suggested that further investigation was warranted. [27] Study protocols were reviewed and approved by the Penn State University Institutional Review Board (IRB # STUDY00016054). Written informed consent was obtained from all participants. All research was conducted in accordance with the Declaration of Helsinki, Good Clinical Practice guidelines and Penn State Health local regulatory requirements.

The study followed the Consolidated Standards of Reporting Trials (CONSORT) reporting guideline and was registered at www.clinicaltrssials.gov under the identification number NCT05697497.

### Setting and participants

Eligible participants were identified through mailing recruitment letters to Penn State Health patients ≥ 65 years of age meeting eligibility criteria and residing in zip codes within a 30-mile radius of Penn State Health-Milton S. Hershey Medical Center. Federal programs like Medicare as well as for the National Institute for Aging (NIA) and Centers for Disease

Control and Prevention (CDC) define "older adults" as people aged 65 or older which informed our decision to include participants 65 years and older.

A total of 415 people were assessed for eligibility between May to September 2021. Eligibility criteria for the trial included being inactive (less than 60 minutes of self-reported physical activity per week and the performance of no RT), reporting difficulty walking(6), 65 years and older, English-speaking, access to the Internet, reporting no chest pain on the PAR-Q, a standard pre-participation risk screener [28] indicating no high risk symptoms for cardiac events during exercise, and correctly completing all questions on the Callahan dementia screener [29]. Of the 415 people screened, 138 participants (33%) were deemed eligible to participate. Of those, 102 participants (74%) were enrolled in the study. Inability to walk without equipment (134 individuals, 48%) was the primary reason for ineligibility for the study, highlighting the need for interventions to improve mobility disability in the elderly population. Prior to randomization, five participants (5%) withdrew, leaving 97 total participants to be randomized. Participant recruitment, delivery of the intervention and measurement of outcomes were conducted between May 1st, 2021, and January 6th, 2022. Fig 1 shows the flow of participants through the trial.

### Randomization and blinding

Participants were randomized through REDCap using stratified assignment based on biological sex and age (65–72 and 73+) to maintain equal representation of older vs. younger males and females in each group. Functional walking limitations rise sharply with age. (2) We typically observe that the median age of participants who enroll in our trials is 72–73 years. For that reason, we selected this as our cut-point on which to stratify randomization. Once randomized, patients were informed of their assigned condition, and their study visits were scheduled. Both participants and the research staff who conducted assessments were not blinded to the participants' treatment assignments. Secondary coders that reviewed videos weekly to assess quality and for safety control were blinded.

### Intervention

Participants were provided with a set of four resistance bands with handles (10–40 pounds of resistance) and a standard aerobics stepper that could be adjusted to 4, 6, or 8 inches in height prior to the baseline visit. They were instructed to perform four exercises daily, each lasting 30 seconds, and to perform as many repetitions as possible during those 30 seconds. Thirty seconds of rest was allowed in between the exercises. Each day participants were asked to perform push-ups, chair stands, two-arm rows, and stair stepping in the same order during each session. No written instructions were provided, although links to videos of each exercise were included in email communications. Participants were instructed to perform the exercises all seven days of the week. Modifications were provided based on the participants' functional level and form.

Push-ups could be modified in any of the following ways: by resting on the knees instead of the toes, by placing the hands on the kitchen countertop (30 inches in most homes), by placing the hands on a set of steps (starting with the 4th step from the floor) or by placing the hands on the wall. When participants were able to perform 15 repetitions of push-ups using one of the modified methods, they were asked to progress to a higher level of difficulty (e.g., place the hands on the 3rd step from the floor).

Chair stands were to be performed, by default, by placing the arms across the chest, but participants could modify them if needed by placing two hands on the knees, one hand on a knee or with arms not placed across the chest. Participants were instructed to use a standard chair with arms, without wheels and with a seat height of approximately 17 inches regardless of the height of the subject. Participants were encouraged to progress, as they were able to do so safely, to the default position of placing their arms across the chest.

Two-arm seated band rows were performed with the resistance band looped around the arches of the feet to allow for a full range of motion. Participants were instructed to pull their elbows back to touch the body to complete a repetition while

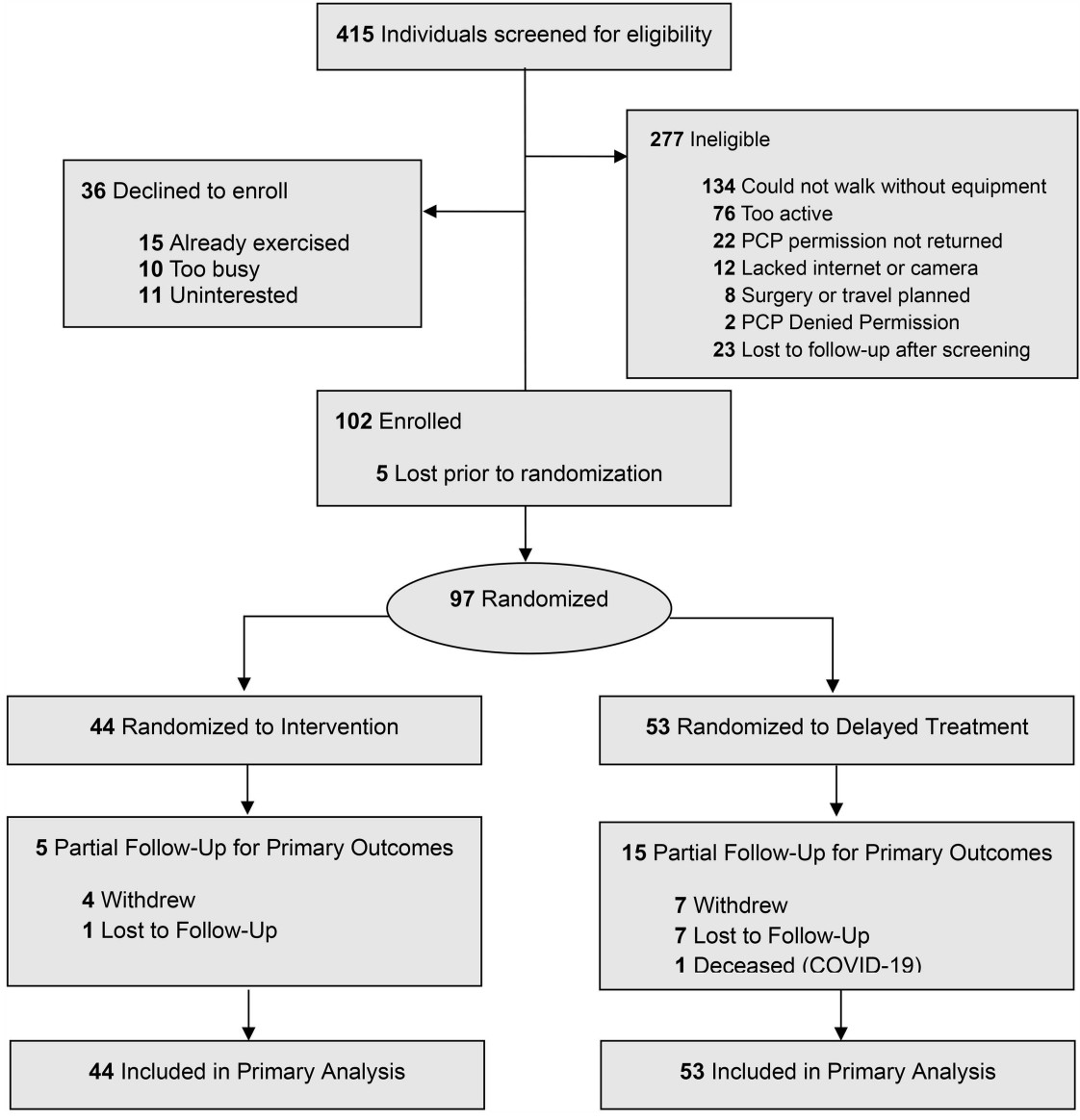

**Fig 1. Participant flow.**

squeezing the shoulder blades behind together. The feet remained flat on the floor to prevent the band from rolling out from under the feet and to prevent injury.

Participants completed the stair stepping exercise by placing their feet up on the step reciprocally and then stepping backwards down in the same fashion. All participants started on the 4-inch step until they could rise fully on the step with both feet six times in 30 seconds. They were then instructed to increase the step height to 6 inches and later to 8 inches.

To increase safety, participants were instructed to place the chair on carpet or against a sturdy object like a wall. For the stair stepping exercise, participants were instructed to place the stair stepper in the corner of a room, so they could touch the wall if they felt unsteady.

**Coaching.** At baseline and at weeks 2, 4 and 8, each participant completed a one-on-one Zoom audio-video coaching session led by a research staff member with expertise in exercise and health psychology with the goal of improving each participant's personal performance records. Each coaching session lasted 10–20 minutes, depending on the participants' instructional needs. The coach began each session by asking the participant a question designed to build rapport (i.e., "small talk"). Next, the coach reviewed the self-reported adherence and performance data of participants, congratulating them on any personal records and positive performance trends. The coach emphasized to the participants the importance of moving as rapidly as possible (without sacrificing form) to pursue a standardized goal of increasing chair stands and stair steps, by four and two repetitions, respectively over the course of the 12-week intervention. The same goal was communicated by the coach to all participants. The stated rationale of this goal was to improve walking ability, as all participants reported difficulty walking at baseline. Before beginning the exercises, the participants oriented their camera so that the entire body was visible by the coach. During each session, the coach passively observed the participant complete the entire workout. The coach asked each participant to complete the workout normally, as if they were not being observed. Upon completing the 4-minute exercise session, the coaches provided feedback and correction (if necessary) relating to exercise form. Where form correction was necessary, the coach demonstrated the proper movement and confirmed understanding by viewing the participant perform the corrected exercise. Modifications and progression methods were suggested and discussed. For example, once a participant could complete 15 push-ups with their hands on a kitchen countertop, the coach encouraged the participant to progress to push-ups with their hands on the 4th step of a staircase (if available in the home of the participant). Safety was emphasized during each session; coaches encouraged participants to place the stepper in a corner (to allow them to regain their balance, if needed) and to do chair stands on a carpeted floor, so the chairs would not slide. The coach ended each session by answering any questions.

**Self-monitoring, feedback, and messaging.** Each participant received an email reminder every morning to complete the daily workout along with a link to a REDCap survey to report their performance on the four exercises (number of push-ups, chair stands, rows and steps on the stepper), whether they used exercise modifications, and their perception of exertion during the workout. The perception of effort was recorded using the 10-point Category-Ratio scale (CR-10) [30]. Participants were only asked to report the rate of perceived exertion (RPE) for the chair stands to minimize participant burden. CR-10 scores less than 5 are considered moderate-intensity, with approximate changes in heart rate of 40–60 beats-per-minute. [31] The self-monitoring form also included an audio file (with timer) to instruct participants through the exercises. In that way it was less of an "extra" thing to do and more of an integral part of the intervention. The survey was brief, lasting less than 5 minutes each day. Emails also included links to videos of each exercise. Every other week, participants received a separate email summarizing their performance and adherence and recognizing their new personal best performance.

## Delayed intervention control

Participants assigned to the delayed intervention control group continued care as usual. Following the completion of the 12-week follow-up, participants in the delayed group began 12 weeks of the physical activity intervention with a one-on-one audio-video Zoom coaching session occurring during the initial first week. At that time, the information given to the physical activity treatment intervention group regarding the daily exercise was given to the delayed group.

## Outcomes

**Functional performance.** The primary outcome was functional performance. The participants in both groups completed three functional performance measures while being observed by a trained researcher over Zoom at baseline, week 6 and week 12 (The Five-Times Sit-to-Stand (FTSTS) test, part of the Short Performance Physical Battery(SPPB) [32,33], the One-Legged Stance Test (OLST) [34] and the 30 second chair stand test [35]. As with the coaching sessions, the participant oriented their camera so that the entire body was visible to the researcher.

The Five-Times Sit-to-Stand (FTSTS) test, part of the Short Performance Physical Battery [32,33] assesses lower extremity strength. Participants were instructed to sit in and fully rise from a chair five times as quickly as possible, without using their arms for support. The test which was timed using a stopwatch ended when the participant's body touched the chair following the fifth repetition. In the One-Legged Stance Test (OLST) subjects were instructed to start with a comfortable base of support, with both eyes open and arms by their side and then stand unassisted on the right leg. The OLST was measured from the time that the left foot was lifted from the floor to when it touched the ground or the other leg [34]. Separately, subjects completed the 30 second chair stand test [35] where participants were instructed to sit in and fully rise from a chair as many times as they could in 30 seconds.

The OLST was performed after the FTSTS and before the 30 second chair stand test, to reduce lower extremity fatigue. The order of tests was consistent between each measurement time point.

Videos were reviewed weekly by a blinded secondary coder to measure quality and safety control for the exercises performed. All timed outcome measure performances were immediately recorded during the Zoom session at the conclusion of each functional performance measure and re-evaluated offline by a blinded reviewer. Discrepancies in recorded times that were greater than one second were replaced with the time recorded by the secondary reviewer. If discrepancies in recorded times were less than 1 second, the average of the two values was used. If there was a disagreement on the number of repetitions performed between the blinded secondary coder and the unblinded research staff member conducting the exercise session, the blinded staff member provided the final adjudication.

**Adherence and retention.** An exercise session was determined to have been completed only if the participants completed the daily REDCap survey reporting their performance, exercise modification use and difficulty level.

If participants did not complete their daily exercise sessions for ≥ 3 days, research staff attempted to contact the lapsed participants via phone and email up to 5 times to re-engage them. Participants were contacted every 48 hours until 14 days of missed daily participation. Participants who did not respond to the project staff for ≥ 14 days were treated as unable-to-be-contacted with no further contact attempts made unless the participant reestablished contact on their own.

**Adverse events.** Participants could report adverse events (AE) to the study team unprompted at any time through the daily exercise survey which asked every participant if they had gotten hurt or injured while participating in the project. Additionally, participants were questioned about injuries explicitly during the scheduled coaching sessions. If a participant answered yes, it prompted them to complete a self-reported injury [36] questionnaire. Each AE was adjudicated for severity and relationship to the study procedures by a sports medicine physician. Given the orthopedic nature of the anticipated AEs, locations of injuries were asked separately to allow for the possibility that more than one body part could have been injured.

## Sample size determination

Effect size calculations are based on evidence that 30 second chair stand performance declines with age and predicts future functional decline. We considered a large effect size as healthier patients in our pilot showed large gains (i.e., $d > 1.00$) in squat performance, but because our intended sample has at least some degree of limitations with walking, we expect a more conservative effect of the intervention on measures of physical function (e.g., $d = .60$).

Based on a 2-group design (intervention, control) assuming a moderate standardized between-group difference at 12 weeks (Cohen's $d = .60$), a sample size of 88 (44 per group) would be needed to detect a difference between groups with a two-sided significance level of 0.05 and 80% power. To account for an anticipated 20% attrition rate over the study period, we will recruit up to 110 participants.

## Statistical analysis

Summary statistics (e.g., mean and standard deviation (SD) for continuous variables; frequency and proportion for categorical variables) were reported for participant baseline characteristics. Baseline differences between randomly assigned

participants were evaluated with two-sample t-tests. Changes in the primary outcome measures over 12 weeks were evaluated at the group level using linear mixed-effect models to analyze change over time and the Group x Time interaction. The analysis employed an intention-to-treat approach, with missing data addressed through mixed-effects modeling in the longitudinal data analysis framework. The linear mixed-effects model assumes linearity, normally distributed residuals and random effects with constant residual variance. These assumptions were evaluated using residual-versus-fitted plots and Q-Q plots of residuals and random-effects estimates, with the adequacy of the random-effects structure examined by comparing nested models using likelihood ratio tests. No meaningful violations were observed. All statistical analyses were performed using statistical software R v4.2.2 with packages tidyverse v1.3.2, nlme v3.1.160, lme4 v1.1.31, and effsize v0.8.1. A two-sided $p$-value of less than or equal to .05 was considered statistically significant.

## Results

### Demographic characteristics

Ninety-seven total participants were randomized to either receive the physical activity treatment intervention (n = 44) or the delayed treatment control condition (n = 53). Participant characteristics are presented in Table 1. Overall, average participant age was 74 years (SD = 6.0) and 68% were female. At baseline, all participants reported that they had difficulty walking a quarter of a mile by themselves, without special equipment. On average, participants reported performing approximately 18 minutes of total light, moderate and vigorous physical activity per week at baseline, far below the recommended guidelines of minimum 150 minutes of moderate intensity or 75 minutes of vigorous intensity physical activity per week for older adults ≥ 65 years old [37,38].

### Intervention effects on functional performance

Fig 2 shows the changes in the functional performance measures over time. At baseline, the mean FTSTS time for the intervention group was 12.2 (4.6) seconds, and for the control group it was 14.4 (5.3) seconds ($p$ = 0.08). After 12 weeks, the FTSTS time of the intervention group was reduced by 2.76 seconds ($p$ < 0.001), whereas the time reduction in the control group was 0.48 seconds ($p$ = 0.49). Overall, the intervention group decreased their FTSTS time by 2.28 seconds more than the control group over 12 weeks (*95% CI: 0.47–4.09;* p = 0.01). There was 83% agreement (<1 second difference) of FTSTS times between live scoring and scoring reviewed offline by the secondary rater.

At baseline, the mean number of chair stands in the intervention group was 9.5 (3.7) and in the control group was 10 (4.3) ($p$ = 0.53). Over 12 weeks, the intervention group increased the number of chair stands completed in 30 seconds by 5.08 repetitions ($p$ < 0.001), whereas the control group increased 0.87 repetitions ($p$ = 0.10). Overall, the intervention group increased the number of chair stands by 4.22 repetitions (95% CI: 2.78–5.66) more than the control group at 12 weeks

**Table 1. Participant characteristics at baseline.**

| Characteristics and Outcomes (%, mean) | Overall (N = 97) Mean (SD), % | Control (N = 53) Mean (SD), % | Intervention (N = 44) Mean (SD), % | Mean difference (Intervention -Control) | p-value |
|---|---|---|---|---|---|
| **Characteristics** | | | | | |
| Age | 74.3 (6.0) | 74.3 (6.2) | 74.4 (5.9) | 0.08 (−2.52,2.35) | 0.95 |
| Gender, female | 66 (68%) | 39 (74%) | 27 (61%) | −12.2% (−8.5%, 33.0%) | 0.20 |
| Total PA, minutes/week | 17.9 (19.5) | 15.9 (19.3) | 20.3 (19.7) | 4.40 (−12.30, 3.50) | 0.27 |
| **Primary Outcomes at Baseline** | | | | | |
| 5 Times Sit-to-Stand, seconds | 13.4 (5.1) | 14.4 (5.3) | 12.2 (4.6) | −2.21 (−0.23, 4.66) | 0.08 |
| Number of Chair Stands, 30 seconds | 9.8 (4.0) | 10.0 (4.3) | 9.5 (3.7) | −0.53 (−1.11, 2.16) | 0.53 |
| One Leg Stance, seconds | 7.5 (8.1) | 7.7 (8.8) | 7.3 (7.2) | −0.38 (−2.90, 3.66) | 0.82 |

 

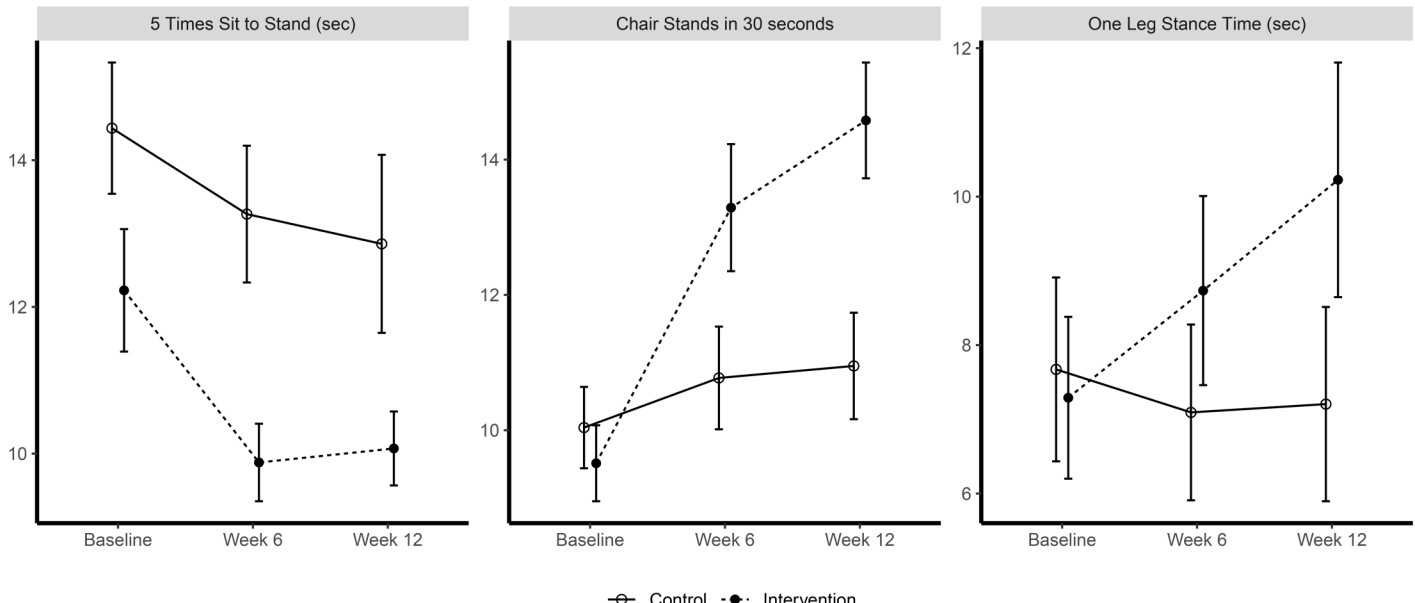

**Fig 2. Changes in Five-Times Sit-to-Stand, 30 second Chair Stand and One-Legged Stance over time.** Error bars represent 95% confidence intervals. Note that the confidence intervals were calculated via the formulae provided by Morey [39].

($p < 0.001$). There was 77% agreement (<1 second difference) of chair stand times between live scoring and scoring reviewed offline by the secondary rater.

The mean OLST time in the intervention was 7.3 (7.2) seconds and in the control group it was 7.7 (8.8) seconds ($p = 0.82$) at baseline. Over 12 weeks, the OLST time increased 2.72 seconds in the intervention group ($p = 0.02$). In the control condition, OLST time decreased 0.85 seconds ($p = 0.45$). Comparing the two groups, the intervention group increased their OLST time by 3.57 seconds (*95% CI: 0.61–6.53, $p = 0.02$*) compared with the control group in 12 weeks. There was 84% agreement (<1 second difference) of chair stand times between live scoring and scoring reviewed offline by the secondary rater.

Table 2 shows the group difference in changes between the intervention and control groups (intervention – control) over the 6- and 12-week periods.

## Fidelity of treatment enactment and retention

Over 12 weeks, participants completed the workout on an average of 81% of days (5.6 days per week), based on an intent-to-treat approach. Modifications were used in 40.6% of exercise sessions. The reported frequencies of modifications (among those 40% of sessions) for each exercise were as follows: push-ups 70.3%, chair stands 24.1%, rows 12.0%, stair stepping 22.9%. Among the modifications provided, the most common modifications used were: wall push-ups, using two hands on knees for the chair stands, using lighter resistance bands for the two arm seated band rows, reducing height of the stepper and using a chair to assist with the stair stepping. Over 12 weeks, the average repetitions increased from 8.1 to 17.5 for pushups, from 7.6 to 16.7 for chair stands, from 10.6 to 27.5 for rows and from 8.7 to 15.8 for stair stepping. While the RPE scores for the chair stands increased slightly from 3.2 to 4.6, it was < 5.0 and within moderate intensity scoring on the CR-10 scale. Five participants (11%) in the intervention group and 15 (28%) in the delayed treatment control condition dropped out at any point or were lost at follow-up. This difference was statistically significant ($p = .04$).

**Table2. Group difference between intervention group and control group (intervention – control).**

| | Group Difference in 6-Week Change (mean [95% CI])) | Group Difference in 12-Week Change (mean [95% CI])) |
|---|---|---|
| 5 Times Sit-to-Stand, seconds | −1.14 [−2.05, −0.24] | −2.28 [−4.09, −0.47] |
| Number of Chair Stands, 30 seconds | 2.11 [1.39, 2.83] | 4.22 [2.78, 5.66] |
| One Leg Stance, seconds | 1.79 [0.31, 3.27] | 3.57 [0.61, 6.53] |

## Adverse events

Over 12 weeks, out of 2994 completed sessions in total, 7 AEs were experienced in six participants, identified to have definite, probable or possible relation to the intervention. These represent an incidence rate of 1 AE per 427 sessions of exercise. All AEs were orthopedic in nature, with shoulder (2/6) discomfort being most reported. Most (6/7) of the AEs led participants to miss one or more workouts. Half (3/6) resulted in the participant visiting a healthcare professional, with one overnight admission for observation in a female participant with shoulder pain, to rule out myocardial infarction.

## Discussion

In this randomized trial of older adults with pre-existing walking difficulty, 12 weeks of a 4-minute brief, home based, daily, functional RT program called FAST (*Functional Activity Strength Training)*-2, which included only 60 seconds of lower extremity RT exercise yielded significant improvement in functional performance compared to usual care. Moderate to large differences between groups were observed, with the RT group improving by 4.2 more repetitions than the Control group in the 30 second chair stand performance (95% CI: 2.8–5.7, p < 0.001), 3.6 more seconds more than the Control group in their OLST time (95% CI: 0.6–6.5, p = 0.02) and a decrease of 2.3 seconds relative to the Control group on the FTSTS test (95% CI: 0.5–4.1, p = 0.01). This is clinically significant as the reported Minimum Clinically Important Difference (MCID) for the 30 second chair stand test is ≥ 2 repetitions for predicting improvement in the 6-minute walk test [40] and the MCID for FTSTS is 2.3 seconds. [41] These results are also consistent with findings in RT literature in general, which note that the first few sets of exercise per week lead to 80% of the gains in strength as higher set volumes. [23] Furthermore, the FAST-2 program had a high level of adherence and uptake, with those assigned to the intervention completing the exercises on an average of 81% of days (5.6 days per week). Collectively, these results suggest that brief, home based, functional RT programs could prevent and improve functional limitations in this growing population potentially changing the trajectory of their mobility disability.

Age match normative values for the FTSTS in older adults are well described for community dwelling older adults (with no major walking difficulty). [42] These community norms underestimate expected times in the mobility impaired population of older adults with pre-existing walking difficulty. [43,44] For older adults with pre-existing walking difficulty, FTSTS time around 15–20 seconds is common, but values ≥ 16 seconds are often interpreted as indicating elevated fall risk and need for strengthening/balance interventions. [43,44] In our study, at baseline, the mean FTSTS time for the intervention group was 12.2 (4.6) seconds, and for the control group it was 14.4 (5.3) seconds (p = 0.08). After 12 weeks, the intervention group decreased their FTSTS time by 2.28 seconds more than the control group (95% CI: 0.47–4.09; p = 0.01). Similar to FTSTS, normative values for the 30 second chair stand test are well described for generally healthy, community-dwelling older adults [33,45], but there are no widely accepted separate norms specifically for those with pre-existing walking difficulty and they likely overestimate expected performance for this group. Scores below 10–12 often signal increased fall risk and poorer mobility and older adults with pre-existing gait problems have been reported to have mean scores around 8–11 stands. [33,46] In our study, at baseline, the mean number of chair stands in the intervention

group was 9.5 (3.7) and in the control group was 10 (4.3) (p = 0.53). Over 12 weeks, the intervention group increased the number of chair stands by 4.22 repetitions (95% CI: 2.78–5.66) more than the control group (p < 0.001). For older adults with pre-existing walking difficulty, a OLST time of less than 5–10 seconds generally indicates a significant balance issue and an increased risk for falls. [47,48] In our study, at baseline, the mean OLST time in the intervention was 7.3 (7.2) seconds and in the control group it was 7.7 (8.8) seconds (p = 0.82). Over 12 weeks, the intervention group increased their OLST time by 3.57 seconds (95% CI: 0.61–6.53, p = 0.02) compared with the control group. These results suggest significant gains in functional strength, and mobility in this mobility impaired population.

Our findings have clinical relevance as it's often challenging to find an exercise program that can be easily adopted and well completed in a population that is often hesitant to begin exercise programs even when offered at no extra cost. Difficulty with walking represents the first stage of decline in physical function in older adults. Our study showed a high level of adherence to the physical activity treatment intervention (81%) over 12 weeks. In a systematic review of home-based exercise programs for older adults with mobility difficulty with an average intervention period of 8–12 weeks, the average rate of adherence was found to be only 67% (12 studies). [49,50] Apart from a social contract and accountability from the periodic video supervision, no further incentives were provided for study completion. Despite this, adherence remained high, supporting the feasibility of the intervention.

The study results extend the existing literature by demonstrating results similar to other studies utilizing brief RT programs, although, with a novel, completely remotely delivered intervention and with minimal supervision. In an uncontrolled study Fujita and colleagues, who exposed older adults to just over 2 minutes of chair stands 3 days per week for 3 months, observed an increase of 18% in knee extension torque. [51] Similarly, in a quasi-experimental study of nursing home residents, Slaughter and colleagues observed that older adults who performed 1 set of repeated chair stands on an average of two times per day experienced a significantly slower decline in sit-to-stand performance and significantly better scores in the Functional Independence Measure [52] after 6 months. [53] Importantly, these studies were in-person, one-on-one supervised interventions, while the present study was conducted entirely virtually with approximately 30 minutes per month of remotely delivered synchronous video supervision.

The brief video supervision format was adopted to increase the potential for dissemination and cost-efficiency. To our knowledge, this represents the first attempt to evaluate brief resistance training among older adults with preexisting mobility disability using a virtual format. In addition to efficacy trials, future work to analyze cost-effectiveness of the intervention would be valuable.

## Strengths and limitations

While the study had many strengths, including a randomized design, a population at high risk for future loss of mobility, a cost-efficient and easily disseminable entirely virtual format, high uptake to adherence and a quality control protocol for the three study outcomes, the study had several limitations. First, COVID restrictions curtailed in-person study visits, so the three key outcome measures needed to be collected via video and other gold-standard measures of lower extremity performance (e.g., SPPB, 1 Repetition Maximum, 6 Minute Walk Distance) were not completed. Of note, the SPPB includes the FTSTS, which our study measured, and several balance measures (e.g., tandem stance), while our study included only the OLST. As moderate-large improvements were observed in both the FTSTS and OLST measures, it is quite likely that improvements in SPPB would also have been observed, yet future studies will need to be performed to ensure that FAST improves these gold standard measures. Second, the intervention was only 3 months in duration. While adherence was excellent (81% of days completed), and adherence over 6 months was good (73% of days) in the first FAST study [26], it is possible that older adults will become bored with the same activities over time and, as a result, adherence will decline. Longer studies will be needed to understand whether adherence, as well as performance improvements, persists. Third, as older adults without access to the Internet were excluded, the participants may not be representative of the population at risk. In 2021, 75% of adults over 65 had access to the Internet, but it is unclear what

level of access is observed for even higher ages, as the average age in this study was 74.3. Fortunately, 96% of US adults ≥ 50 use the Internet, so lower rates of Internet access today among older adults are likely only temporary. [54] In the near future, nearly all older adults will likely have Internet access. Fourth, the intention-to-treat principle preserves the benefits of randomization, but ITT estimates may underestimate the true effect of the intervention in the presence of nonadherence. Missing outcome data and loss to follow-up, while addressed using linear mixed-effects models under a missing-at-random assumption, may still bias estimates if missingness depends on unobserved factors. Consequently, ITT results should be interpreted as conservative estimates of intervention effectiveness under real-world implementation rather than as estimates of efficacy under full adherence. Fifth, the sample size for the study was small relative to the prevalence of functional limitations in the elderly population. Larger studies will be needed to clarify the external validity of study findings. Finally, it is important to state clearly what this study does not suggest. It does not suggest that four minutes of daily exercise is sufficient to result in body composition or cardiovascular health improvements for the general population; instead, it only suggests that four minutes of daily exercise was sufficient to improve functional performance in older adults with existing walking difficulties.

## Conclusion

In conclusion, a brief 4-minute daily functional RT program, which included only 60-seconds of lower extremity exercises, showed significant improvement in measures of lower extremity performance that are strongly associated with future disability in older adults. Future studies are needed to understand whether these differences are maintained with gold-standard measures and whether adherence is maintained over a longer period.

## Supporting information

**S1 File. Consort 2025 Checklist.**
(DOCX)

**S2 File. Trial Protocol.**
(DOCX)

## Acknowledgments

We thank the trial participants for their participation in this trial. We are sincerely grateful for the efforts of our research support team; without whose help we would be unable to facilitate this work.

## Author contributions

**Conceptualization:** Matthew A Ladwig, David E Conroy, Kathryn H Schmitz, Christopher Sciamanna.

**Data curation:** Jordan Kurth, Matthew A Ladwig, Shouhao Zhou, Christopher Sciamanna.

**Formal analysis:** Smita Dandekar, Jordan Kurth, Yimeng Shang, David E Conroy, Kathryn H Schmitz, Liza S Rovniak, Shouhao Zhou, Christopher Sciamanna.

**Investigation:** Matthew A Ladwig, Christopher Sciamanna.

**Methodology:** Matthew A Ladwig, David E Conroy, Kathryn H Schmitz, Christopher Sciamanna.

**Project administration:** Jordan Kurth, Matthew A Ladwig, Christopher Sciamanna.

**Resources:** Matthew A Ladwig, Christopher Sciamanna.

**Software:** Yimeng Shang.

**Supervision:** Matthew A Ladwig, Christopher Sciamanna.

**Validation:** Yimeng Shang, Christopher Sciamanna.

**Visualization:** Christopher Sciamanna.

**Writing – original draft:** Smita Dandekar, Jordan Kurth, Yimeng Shang, David E Conroy, Liza S Rovniak, Shouhao Zhou, Christopher Sciamanna.

**Writing – review & editing:** Smita Dandekar, Jordan Kurth, Yimeng Shang, Jonathan G Stine, Matthew A Ladwig, David E Conroy, Kathryn H Schmitz, Liza S Rovniak, Matthew Silvis, Margaret Danilovich, Noel Ballentine, Natalia Pierwola-Gawin, Shouhao Zhou, Christopher Sciamanna.

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
