## [Decision Letter · Decision Letter 0]

22 Nov 2025

Brief daily functional strength training to improve functional performance in older adults with mobility disability: A randomized trial

PLOS ONE

Dear Dr. Dandekar,

Thank you for submitting your manuscript to PLOS ONE. After careful consideration, we feel that it has merit but does not fully meet PLOS ONE’s publication criteria as it currently stands. Therefore, we invite you to submit a revised version of the manuscript that addresses the points raised during the review process.

We look forward to receiving your revised manuscript.

Kind regards,

Leonardo Roever

Academic Editor

PLOS ONE

Journal Requirements:

2. We note that you have selected “Clinical Trial” as your article type. PLOS ONE requires that all clinical trials are registered in an appropriate registry (the WHO list of approved registries is at https://www.who.int/clinical-trials-registry-platform/network/primary-registries " https://www.who.int/clinical-trials-registry-platform/network/primary-registries and more information on trial registration is at http://www.icmje.org/about-icmje/faqs/clinical-trials-registration/ ). Please state the name of the registry and the registration number (e.g. ISRCTN or ClinicalTrials.gov ) in the submission data and on the title page of your manuscript. a) Please provide the complete date range for participant recruitment and follow-up in the methods section of your manuscript. b) If you have not yet registered your trial in an appropriate registry, we now require you to do so and will need confirmation of the trial registry number before we can pass your paper to the next stage of review. Please include in the Methods section of your paper your reasons for not registering this study before enrolment of participants started. Please confirm that all related trials are registered by stating: “The authors confirm that all ongoing and related trials for this drug/intervention are registered”. Please see http://journals.plos.org/plosone/s/submission-guidelines#loc-clinical-trials for our policies on clinical trials.

3. In the online submission form, you indicated that “The data that support the findings of this study are available upon request to the correspondent author”

“The author(s) declared the following potential conflicts of interest with respect to the research, authorship, and/or publication of this article: Dr. Sciamanna is part-owner of BandUp, Inc. and Play Fitness, LLC, formed to test the viability of various business models to disseminate findings from exercise studies. Dr. Jonathan Stine receives or has received research support from Astra Zeneca, Galectin, Kowa, Noom, Inc, Novo Nordisk, Regeneron and Zydus Therapeutics. Dr. Stine consults for Novo Nordisk and is on an advisory board for Madrigal. No other authors have competing interests.”

Additional Editor Comments:

(PLACE INSERTS IN A DIFFERENT COLOR FONT TO IDENTIFY CHANGES IN THE ARTICLE)

PUT IN RED LETTERS - ANSWER THE QUESTIONS BELOW POINT BY POINT.

INCLUDE THE PAGE AND LINE WHERE YOU ARE MAKING THE CHANGE.

PLEASE INCLUDE ALL REQUESTS IN THE MANUSCRIPT.

Include in article

0 - Please correct grammatical and spelling errors

1 - Abstract

Conclusions: State only what your study found; do not include extraneous information not backed up by the results.

2 - Discussion

Compare and contrast your study with others in the most relevant world literature, particularly the recent literature.

3 - What new information is sufficient to modify existing clinical practice?

4 -What are the conclusions and implications for current practice, and particularly for future research that may have a significant impact on clinical decisions?

5 - How can this study affect public policies related to health?

6 - What does this study add to the literature?

7 – Improve - At the end of the Discussion, under the subheading "Limitations," review the limitations of your study.

8 - At the end of the limitations, under the subheading " Future directions".

9 - Conclusion

Take special care to draw your conclusions only from your results and verify that your conclusions are firmly supported by your data.

Reviewer's Responses to Questions

**Comments to the Author**

1. Is the manuscript technically sound, and do the data support the conclusions?

Reviewer #1: Yes

Reviewer #2: Yes

2. Has the statistical analysis been performed appropriately and rigorously?

Reviewer #1: Yes

Reviewer #2: Yes

3. Have the authors made all data underlying the findings in their manuscript fully available?

Reviewer #1: Yes

Reviewer #2: Yes

4. Is the manuscript presented in an intelligible fashion and written in standard English?

Reviewer #1: Yes

Reviewer #2: Yes

Reviewer #1: Dear Editor,

I write to submit my review of the manuscript titled “Brief daily functional strength training to improve functional performance in older adults with mobility disability: A randomized trial.”

The study via randomized trial evaluated the effects of a 12-week brief, home-based functional RT program, FAST (Functional Activity Strength Training)-2, on adherence and functional impairment in older, inactive adults ≥ 65 years of age, with pre-existing walking difficulty.

Comments

1. Kindly explain what informed age stratifiers of 65-72 and 73+ are? Include justification in the manuscript. Is 65 years the official definition of old age in the USA? Why did you consider only those who were greater than 65 years? Why not 60 years, etc? Kindly include justification for the age inclusion threshold.

2. What was the rationale behind the 12-week, delayed-treatment randomized trial? What would have been the effect if the duration had been extended beyond 12 weeks or reduced? The rationale for selecting 12 weeks for implementing the intervention should be included in the manuscript.

3. Major concern: There was no power analysis (sample size calculation) to determine whether the study was powered enough to assess the effectiveness of the intervention if it exists. No properly estimated sample size. The authors initially screened 415 participants and ultimately arrived at 102 for the main study; however, there is no scientific justification for the final sample size of 102. Would the result have been different if only 60 participants had agreed to participate in the study? More rigorous power analysis needs to be conducted to justify the 120 participants

4. For each of the outcome measures, kindly be clear on the measurement scale (continuous, binary, discrete, nominal, ordinal, multinomial outcome) etc. This will determine the appropriateness of the statistical methods employed.

5. The authors stated that “Summary statistics were reported for participant baseline characteristics”. This is too broad. Kindly state the key summary statistics that were reported

6. Include key assumptions of the linear mixed effect model and how these assumptions were tested.

7. Include the limitations of the intention-to-treat estimates

8. Include the mean difference between the intervention and control by creating an additional column before the p-value estimates in Table 1

Reviewer #2: The rationale for study is valid and provides an alternative to current recommendations that are needed for older adults, particularly those with a mobile disability. This FAST-2 program is brief, has appropriate progression, and is easily delivered, with internet access. There was strong adherence to the training program and outcomes show improvements in the training group that will improve the ability to complete ADLs and reduce injury in this population. Methodology is sound and the verification of measured outcomes with a second interpreter is a good added control. A few things should be addressed.

I am concerned about the two-arm seated band rows and the potential for the band to roll up the foot and injure. Did this happen at all in your study? Can you include an alternate exercise targeting similar muscle groups in the discussion as an alternative in case one is not comfortable doing this?

The self-monitoring, feedback, and messaging seems potentially laborious for an older person. Was any data collected on how this what utilized by the subjects or how these affected adherence?

Inclusion of the REDCap survey values, RPEs, and the common exercise modifications the subjects used would be valuable data for inclusion.

PG 13: Either reverse the order of the OLST and the CST in figure2 and table 2, or reverse it in the text so that the order parallels each other.

Include discussion on where the subjects land in relation to norms for the functional tests and how this might affect how much they were able to improve, and generally what the scores for the people at baseline and in the control group translate to.

**Do you want your identity to be public for this peer review?** For information about this choice, including consent withdrawal, please see our Privacy Policy

Reviewer #1: No

Reviewer #2: No

---

## [Author Response · Author response to Decision Letter 1]

22 Dec 2025

We respectfully submit a revised version of our manuscript incorporating changes suggested by you and the reviewers. We have addressed all points raised during the review process.

Please see attached "Response to Reviewers" document along with revised manuscript, revised S2 trial protocol and revised S1 consort 2025 checklist

---

## [Decision Letter · Decision Letter 1]

7 Jan 2026

Brief daily functional strength training to improve functional performance in older adults with mobility disability: A randomized trial

PLOS One

Dear Dr. Dandekar,

Thank you for submitting your manuscript to PLOS ONE. After careful consideration, we feel that it has merit but does not fully meet PLOS ONE’s publication criteria as it currently stands. Therefore, we invite you to submit a revised version of the manuscript that addresses the points raised during the review process.

We look forward to receiving your revised manuscript.

Kind regards,

Leonardo Roever PhD, MBA

Academic Editor

PLOS One

Journal Requirements:

Reviewers' comments:

Reviewer's Responses to Questions

**Comments to the Author**

Reviewer #1: All comments have been addressed

Reviewer #2: (No Response)

2. Is the manuscript technically sound, and do the data support the conclusions?

Reviewer #1: Yes

Reviewer #2: Yes

3. Has the statistical analysis been performed appropriately and rigorously?

Reviewer #1: Yes

Reviewer #2: Yes

4. Have the authors made all data underlying the findings in their manuscript fully available?

Reviewer #1: (No Response)

Reviewer #2: Yes

5. Is the manuscript presented in an intelligible fashion and written in standard English?

Reviewer #1: Yes

Reviewer #2: Yes

Reviewer #1: The authors have comprehensively addressed all the comments and concerns raised in my previous review of the manuscript

Reviewer #2: The rationale for study is valid and provides an alternative to current recommendations that are needed for older adults, particularly those with a mobile disability. This FAST-2 program is brief, has appropriate progression, and is easily delivered, with internet access. However, some reviewer concerns were not adequately addressed and incorporated into the text of the revision. See below

L131-133: The answer to reviewer 1 comment was inadequately incorporated. Add the additional context that's included in the response to reviewers.

L194-197: Add that feet remained flat on floor to prevent rolling of band/ injury.

Section “Self-monitoring, feedback, and messaging” : Include text from response to reviewers that highlight this is not a burden.

Reviewer 2, #3: The purpose of this manuscript is clear, Inclusion of the requested data either as a table or in text aids this and gives more information about the strain on participants, the movement capacity and ability of the participants, and therefore the feasibility to support adoption of this program. This detail is needed to change current practice.

Reviewer 2, #5: I was able to find values and indicators for the FTSTS, the OLST, 30s chair stand test. Not knowing how these participants compare to the population they represent or a related population is a major limitation.

**Do you want your identity to be public for this peer review?** For information about this choice, including consent withdrawal, please see our Privacy Policy

Reviewer #1: No

Reviewer #2: No

---

## [Author Response · Author response to Decision Letter 2]

28 Jan 2026

January 28, 2025,

Dear Editor,

We want to thank you and the reviewers for taking the time to review our revised manuscript titled “Brief daily functional strength training to improve functional performance in older adults with mobility disability: A randomized trial” where we provide experimental evidence that a 12-week (4-minute) , home-based functional resistance training program augmented with goal setting, FAST (Functional Activity Strength Training)-2, can enhance adherence and improve functional impairment in older, inactive adults ≥ 65 years of age, with pre-existing walking difficulty. We greatly appreciate your thoughtful review of our response to the points raised by the reviewers. We are pleased to hear that we adequately addressed the comments raised by reviewer # 1 and are hopeful that this second revision addresses the comments mentioned by reviewer # 2 in the prior revision. The new revisions are mentioned in red in the body of the manuscript.

Please find our detailed responses to recent comments by reviewer # 2 below.

1. L131-133: The answer to reviewer 1 comment was inadequately incorporated. Add the additional context that's included in the response to reviewers.

Response: We thank the reviewer for this thoughtful suggestion and agree that our explanation of the rationale for the 12-week duration of the trial could be improved. We have added additional context as recommended by the reviewer. This section has been updated, and the revised text reads as follows: “This duration was selected as 12 weeks is the typical length of exercise trials. Data suggests that 12 weeks is long enough to allow for physiological changes to occur, both the neural and muscular changes associated with strength training. Initial trials (i.e., those testing a new intervention) typically last 12 weeks to balance participant burden with the ability to test the efficacy of the program. Then, longer trials can be conducted if the 12-week intervention suggested that further investigation was warranted. (Lines 132-137)

2. L194-197: Add that feet remained flat on floor to prevent rolling of band/ injury.

Response: Thank you for your comment. We have added the sentence as suggested by the reviewer. This section has been updated in text as follows: “The feet remained flat on the floor to prevent the band from rolling out from under the feet and to prevent injury.” (Lines 202-203)

3. Section “Self-monitoring, feedback, and messaging”: Include text from response to reviewers that highlight this is not a burden.

Response: We have included text from the response to reviewers in the section “Self-monitoring, feedback, and messaging” that highlights it was not a burden. The text has been modified and reads as follows: “The self-monitoring form also included an audio file (with timer) to instruct participants through the exercises. In that way it was less of an “extra” thing to do and more of an integral part of the intervention. The survey was brief, lasting less than 5 minutes each day. Emails also included links to videos of each exercise.” (Lines 245-248)

4. Reviewer 2, #3: The purpose of this manuscript is clear, Inclusion of the requested data either as a table or in text aids this and gives more information about the strain on participants, the movement capacity and ability of the participants, and therefore the feasibility to support adoption of this program. This detail is needed to change current practice.

Response: We thank the reviewer for this comment. While the purpose of this manuscript is to report the effects of a 12-week brief, home-based functional RT program, FAST (Functional Activity Strength Training)-2, on functional impairment (the primary outcome) and adherence in older, inactive adults ≥ 65 years of age, with pre-existing walking difficulty, we agree that Inclusion of the common exercise modifications used by the participants and information about REDCap survey values on the reps and RPEs (rate of perceived exertion) is helpful information supporting the feasibility of adoption of the program. We have included information on the RPE scale used in the Materials and Methods section which reads as follows: “The perception of effort was recorded using the 10-point Category-Ratio scale (CR-10)(30). Participants were only asked to report the rate of perceived exertion (RPE) for the chair stands to minimize participant burden. CR-10 scores less than 5 are considered moderate-intensity, with approximate changes in heart rate of 40–60 beats-per-minute. (31)” (Lines 241-245).

We have now also included information about the modifications used in the exercise sessions, information on the REDCap survey values on repetitions and the RPEs. The included text now reads “Modifications were used in 40.6% of exercise sessions. The reported frequencies of modifications (among those 40% of sessions) for each exercise were as follows: push-ups 70.3%, chair stands 24.1%, rows 12.0%, stair stepping 22.9%. Among the modifications provided, the most common modifications used were: wall pushups, using two hands on knees for the chair stands, using lighter resistance bands for the two arm seated band rows, reducing height of the stepper and using a chair to assist with the stair stepping. Over 12 weeks, the average repetitions increased from 8.1 to 17.5 for pushups, from 7.6 to 16.7 for chair stands, from 10.6 to 27.5 for rows and from 8.7 to 15.8 for stair stepping. While the RPE scores for the chair stands increased slightly from 3.2 to 4.6, it was < 5.0 and within moderate intensity scoring on the CR-10 scale.” (Lines 381-390)

5. Reviewer 2, #5: I was able to find values and indicators for the FTSTS, the OLST, 30s chair stand test. Not knowing how these participants compare to the population they represent or a related population is a major limitation.

Response: We thank the reviewer for this insightful suggestion. We acknowledge that not knowing how the performance times for the FTSTS, OLT and 30 second chair stand test of the participants compared to the related population is a limitation. We have modified the text to include an entire paragraph with information about relative normative values. Additionally, as these “normative” values as not well established for older adults with pre-existing walking difficulty, we have included minimum clinically important differences (MCIDs)– i.e., thresholds for change that indicate a clinically important improvement. We are hopeful that these additions in the “Discussion” section meet the reviewer’s expectations. The text now reads as below:

“Age match normative values for the FTSTS in older adults are well described for community dwelling older adults (with no major walking difficulty). (42) These community norms underestimate expected times in the mobility impaired population of older adults with pre-existing walking difficulty. (43, 44) For older adults with pre-existing walking difficulty, FTSTS time around 15-20 seconds is common, but values ≥16 seconds are often interpreted as indicating elevated fall risk and need for strengthening/balance interventions.(43, 44) In our study, at baseline, the mean FTSTS time for the intervention group was 12.2 (4.6) seconds, and for the control group it was 14.4 (5.3) seconds (p = 0.08). After 12 weeks, the intervention group decreased their FTSTS time by 2.28 seconds more than the control group (95% CI: 0.47-4.09; p = 0.01). Similar to FTSTS, normative values for the 30 second chair stand test are well described for generally healthy, community dwelling older adults (33)(45), but there are no widely accepted separate norms specifically for those with pre existing walking difficulty and they likely overestimate expected performance for this group. Scores below 10-12 often signal increased fall risk and poorer mobility and older adults with pre existing gait problems have been reported to have mean scores around 8–11 stands.(33)(46) In our study, at baseline, the mean number of chair stands in the intervention group was 9.5 (3.7) and in the control group was 10 (4.3) (p = 0.53). Over 12 weeks, the intervention group increased the number of chair stands by 4.22 repetitions (95% CI: 2.78-5.66) more than the control group (p < 0.001). For older adults with pre-existing walking difficulty, a OLST time of less than 5 to 10 seconds generally indicates a significant balance issue and an increased risk for falls. (47, 48) In our study, at baseline, the mean OLST time in the intervention was 7.3 (7.2) seconds and in the control group it was 7.7 (8.8) seconds (p = 0.82). Over 12 weeks, the intervention group increased their OLST time by 3.57 seconds (95% CI: 0.61-6.53, p = 0.02) compared with the control group. These results suggest significant gains in functional strength, and mobility in this mobility impaired population.” (Lines 419-442)

---

## [Decision Letter · Decision Letter 2]

23 Feb 2026

Brief daily functional strength training to improve functional performance in older adults with mobility disability: A randomized trial

PONE-D-25-57116R2

Dear Dr. Dandekar,

We’re pleased to inform you that your manuscript has been judged scientifically suitable for publication and will be formally accepted for publication once it meets all outstanding technical requirements.

Kind regards,

Domiziano Tarantino, MD

Academic Editor

PLOS One

Additional Editor Comments (optional):

Reviewers' comments:

Reviewer's Responses to Questions

**Comments to the Author**

Reviewer #1: All comments have been addressed

Reviewer #2: All comments have been addressed

2. Is the manuscript technically sound, and do the data support the conclusions?

Reviewer #1: Yes

Reviewer #2: Yes

3. Has the statistical analysis been performed appropriately and rigorously?

Reviewer #1: Yes

Reviewer #2: Yes

4. Have the authors made all data underlying the findings in their manuscript fully available?

Reviewer #1: Yes

Reviewer #2: Yes

5. Is the manuscript presented in an intelligible fashion and written in standard English?

Reviewer #1: Yes

Reviewer #2: Yes

Reviewer #1: The authors have comprehensively addressed all the comments and concerns raised in my previous review of the manuscript

Reviewer #2: The authors have carefully considered previous comments. Concerns have been comprehensively and adequately addressed.

**Do you want your identity to be public for this peer review?** For information about this choice, including consent withdrawal, please see our Privacy Policy

Reviewer #1: No

Reviewer #2: No

---

## [Editor Report · Acceptance letter]

PONE-D-25-57116R2

PLOS One

Dear Dr. Dandekar,

I'm pleased to inform you that your manuscript has been deemed suitable for publication in PLOS One. Congratulations! Your manuscript is now being handed over to our production team.

Kind regards,

on behalf of

Dr. Domiziano Tarantino

Academic Editor

PLOS One